# Developing the Chinese Academic Map Publishing Platform

**Yongming Xu** [1] , **Benjamin Lewis** [2] **and Weihe Wendy Guan** [2],*

[1]  School of Humanities, Zhejiang University, Hangzhou 310027, China; yongmingxu1967@zju.edu.cn
[2]  Center for Geographic Analysis, Harvard University, Cambridge, MA 02138, USA; blewis@cga.harvard.edu
*   Correspondence: wguan@cga.harvard.edu; Tel.: +1-617-496-6102

**Abstract:** The discipline of the humanities has long been inseparable from the exploration of space and time. With the rapid advancement of digitization, databases, and data science, humanities research is making greater use of quantitative spatiotemporal analysis and visualization. In response to this trend, our team developed the Chinese academic map publishing platform (AMAP) with the aim of supporting the digital humanities from a Chinese perspective. In compiling materials mined from China's historical records, AMAP attempts to reconstruct the geographical distribution of entities including people, activities, and events, using places to connect these historical objects through time. This project marks the beginning of the development of a comprehensive database and visualization system to support humanities scholarship in China, and aims to facilitate the accumulation of spatiotemporal datasets, support multi-faceted queries, and provide integrated visualization tools. The software itself is built on Harvard's WorldMap codebase, with enhancements which include improved support for Asian projections, support for Chinese encodings, the ability to handle long text attributes, feature level search, and mobile application support. The goal of AMAP is to make Chinese historical data more accessible, while cultivating collaborative opensource software development.

**Keywords:** Chinese; humanities; history; mapping; platform; opensource; AMAP; WorldMap

## 1. Introduction

Preserved in the vast ancient and modern literature of China, embedded in its unequaled record of human civilization over many millennia and across great regions of land and sea, there exists a vast amount of geographically referenced historic information. Thanks to recent technological advancements, we now have a chance to improve our understanding of the history of this part of the world through the lens of geographic information systems (GIS). We suggest that there exists an opportunity to make geographic relationships more explicit, especially those relating to human activities. At the level of an individual, this may include the geographical distribution of his/her place of origin, travel routes, and social relations. For a group, this may involve the group's distribution and migration trajectory. For non-living objects, this may represent their distribution and change. China contains a holistic collection of geographic information about its people, things, and events throughout history, and we aim to make this information more explicit and usable for scholars. To provide an online place for such materials to be gathered and shared, we developed the Chinese Academic Map Publishing Platform (AMAP).

### 1.1. An overview of Chinese Historical Research

For many thousands of historical figures in China, their birthplace, activities, titles or positions, social networks, and artistic output such as poems, inscriptions, and writings have been recorded,

including geographical references for most figures. For geographic places, there are countless features with names on the landscape, such as administrative villages, towns, neighborhoods, gates, stations, passes, mountains, lakes, springs, waterfalls, bridges, caves, marshes, ponds, etc. For built structures, there are records describing palaces, pavilions, temples, private houses, and other features of the built environment. Even for a significant portion of events taking place thousands of years ago, geographic location information has been preserved.

The reason this record exists is that through the centuries, Chinese scholars have always paid great attention to the recording and studying of geographic relationships. "Yu Gong" (禹贡), China's oldest book of geography, recorded mountains, rivers, terrain, soils, and their properties. "The Geography" (地理志) that first appeared in "Han Shu" (汉书) became an indispensable part of the official history of future generations of Chinese scholars [1]. Local history was recorded in local history books (地方志), and these books form a rich set of local geographic information in China. In addition, there are many other publications about geography in China. In the Confucian "Classics, History, Philosophy, and Literature," (经史子集) four categories of publication, the "geographical" bibliography in the History category, cataloged ancient geography publications. For example, "Yuanhe County Map" (元和郡县图志), "Tai Ping Huan Yu Ji" (太平寰宇记), "Yuan Feng Jiu Ji Zhi" (元丰九域志), "Da Ming Yi Tong Zhi" (大明一统志), "Da Qing Yi Tong Zhi" (大清一统志) and "Du Shi Fang Ji Ji Yao" (读史方舆纪要) are all important works of geography. In addition to textual records and research writings, the ancients also used maps to mark territories and place names, calling them yutu (舆图) or illustration maps. This type of illustration map should be thought of as a schematic diagram rather than as a modern map drawn to scale. A wealth of geographic information describing ancient China is stored in such illustrations.

There are many ways in which an understanding of geographic relationships between people and places is important for understanding the history of China. These include connections between people, events, customs, products, and poetry [2]. Traditional Chinese illustration maps (舆图), are an important source of historical geographic information, however there are some disadvantages to these traditional geographic renderings. First, it is difficult to use such maps as-is for spatial analysis. For example, when we compile a local chronical gazetteer of a county, the local writers and editors may know the county's population and local elites well. However, it is difficult for them to know the relationship between the population and elites at the national level. In order to show the distribution of historical actors across the country, one would need to gather and map hundreds of thousands, or even millions of data points. In addition, although traditional maps can be visualized and labeled, locations are usually not accurate. Finally, paper maps are hard to access, limiting their audience. Modern geographic information systems (GIS), in contrast, solved some of these problems and enabled digital maps to spread quickly.

*1.2. The Need for a Platform*

The widespread digitization of historical materials in the 21st century has made computers central to the work of scholars. As the volume of digital historic documents increases, so has the need to make this content available for scholars to explore and analyze via the web. This has led to improvements in platforms for storing and managing such content [3]. Looking ahead, given the large volume of structured, digitized historical content available, a range of research opportunities now exist for applying machine learning and other processing methods to the analysis of such collections.

Geospatial data describing historical events and people have many uses, but there are few platforms that are designed to handle the unique characteristics of historical geographic data, as relates to data storage query, display, and analysis [4]. Knowing where something happened is important for understanding why it happened, and for seeing relationships between events that might otherwise go unnoticed. In addition, for a historically oriented system, knowing when something happened is important. There exist even fewer platforms which support basic temporal search or display capabilities in addition to spatial capabilities.

Geospatial technology is being used in a variety of sectors, including government, military, commercial, non-profit, and academic organizations, which has resulted in the creation of a large volume of well-structured geospatial information covering many topic areas. For researchers in any discipline to create and share such data, digital mapping platforms are required. In a humanities context, such a platform should support the following core capabilities:

- Use by a distributed group of scholars from many institutions to upload and create new spatial datasets online, compose maps, symbolize layers, and comment on each other's layers.
- Standard GIS map exploration tools such as pan, zoom, identify
- Metadata creation tools
- Ability to search for content by time, space, and keyword
- Ability to search for content on local or remote servers
- Allow users to control access to content they own.
- Support multiple languages for the user interface

There are many online mapping platforms in existence [5], most of which are developed and used by scholars outside China. Scholars inside China need a platform specifically developed with China's unique Internet environment in mind, which supports appropriate map projections, language encodings, and other characteristics important for Chinese users.

## 2. An Overview of Existing Platforms for Historical GIS and Digital Humanities

There exist a wide variety of online platforms which provide scholars with access to historical spatial information in the context of the digital humanities, a few of which we have listed below. Each of these platforms present ideas for developing a better system to enable digital humanities data to be generated, brought together, and made available to scholarly communities across China. The systems we looked at fall into three broad categories:

The first category provides access to a particular data collection. These systems are library-like in that they represent authoritative datasets, which are well curated and generally do not change. Typically, the data are created and maintained by one organization on a particular subject matter. These systems do not provide tools for outside users to create or edit data on the site. Examples of such systems include:

- **China Biographical Database (CBDB)** This project (中国历代人物传记数据库 in Chinese) is led by Professor Peter K. Bol of Harvard University. The collaborators include the Research Center for Ancient Chinese History of Peking University and the Institute of History and Language of the Central Research Institute of Taiwan. The Chinese Historical Biography Database is currently the world's largest database of Chinese historical figures' biographical data for analysis. Some 400,000 Chinese historical figures are recorded in it, and it contains nearly 500,000 people from other sources, such as Chinese local chronicle gazetteers. The database is made freely accessible for online query and download [6].
- **China Historical GIS (CHGIS)** This database provides data on the historical political divisions of China. The project is also led by Professor Peter K. Bol of Harvard University with project management by Lex Berman. This project is in collaboration with Fudan University's Center for Historical Geographical Studies to vectorize Chinese historical place names and historical maps. The database also records administrative hierarchies of place names and their evolution through time in the form of a relational database. The database is accessible for online query and may also be downloaded for free [7,8].
- **Great Britain Historical Geographical Information System (GBHGIS)** This project documents the changing human geography of the British Isles since the first census in 1801 and is developed and hosted by the University of Portsmouth. It was sponsored by the UK government [9].
- **U.S. National Historical Geographic Information System (NHGIS)** This archive provides access to census data for the United States back to 1790. The National Historical Geographic Information

System is developed by the Minnesota Population Center (MPC) and sponsored by US federal funding [10].

- **Rumsey Map Collection** This collection contains over 150,000 maps and related images at a variety of scales in global locations, ranging in date from about 1550 to the present. The Rumsey Map Collection is hosted by Stanford University and sponsored by philanthropist David Rumsey. This collection focuses on rare 16–21st century maps of North and South America, as well as maps of the World, Asia, Africa, Europe, and Oceania. The collection includes atlases, wall maps, globes, school geographies, pocket maps, books of exploration, maritime charts, and a variety of cartographic materials, including pocket, wall, children's, and manuscript maps. Items range in date from about 1550 to the present [11].

These are just a few of the existing archives of historical spatial data. Other systems include the Belgian historical GIS project at Ghent University, Belgium, the Batanes Islands Cultural Atlas developed by the University of California at Berkeley, the New York City Historical GIS Project developed by the New York City Public Library, and the Historical Geographic Information System developed by Wikipedia, among others. These systems provide authoritative source material for scholars and are often based outside of China. This situation points to an important goal of AMAP: to provide a system oriented toward Chinese scholars in particular, and which supports the creation of new information as well as the use and interpretation of existing materials.

A second category of online GIS system enables users to search across multiple distributed collections. This type of system, or registry of systems, harvests metadata from multiple federated collections and stores them in a central registry to make data easier to discover. Examples of this type of system include:

- **Geoplatform** This system is developed by the member agencies of the U.S. Federal Geographic Data Committee, and maintains a registry of over 160,000 datasets that are distributed across many state and government agency portals within the United States [12].
- **INSPIRE Geoportal** The INSPIRE Geoportal is the central access point for spatial data provided by European Union member states. The Geoportal maintains a registry of datasets that are distributed across many national geospatial portals [13].
- **Harvard Geospatial Library** This system contains historic map images, census data, place names and locations, road networks, elevation models, and many other layers distributed across the libraries of many universities [14].
- **Old Maps Online** This system indexes over 400,000 historic scanned maps that are served up from many libraries and archives [15].
- **Digital Public Library of America** This system maintains a registry of over 34 million maps, photographs, books, news footage, oral histories, personal letters, museum objects, and artwork that are distributed across hundreds of cultural institutions in the United States [16].

The data search and discovery capabilities in these systems enable access to data distributed across many systems. This provided ideas for the design of AMAP. If one were to choose any data and ask where it can be found, the answer will usually include many systems. Therefore, for a scholar, knowing where to look is half the battle. An architecture capable of harvesting metadata from many systems and presenting it together in a central registry should make specialized data easier to find.

A third category of online GIS systems is a hybrid system which allows users to search for content as described above but also allows them to create and edit content directly online which can then be made available for others to discover as a dynamic library. This kind of system combines data search tools with additional tools for data and metadata creation, as required by the AMAP project. Some of these systems include a registry, as described in the previous category, to enable users to find and use data which resides on other systems in addition to local data. Below, we will examine the hybrid systems from the perspective of fitness as a starting point for AMAP development.

- **ArcGIS Online** This platform is a full featured commercial hosted product developed by the Environmental Systems Research Institute (Esri). This platform is comprised of applications and templates for creating and sharing interactive maps and is an integral part of Esri's ArcGIS suite of applications. The platform supports a wide range of GIS data symbolization, curation, and sharing capabilities. While ArcGIS Online supports the functionality required by the AMAP project, it was not chosen for the project because the AMAP system must be run from inside the Zhejiang University firewall [17].

- **CartoDB** This system is a hosted GIS platform for building online GIS applications. It is especially well suited to business and government settings in which a developer builds custom applications for the organization using the Carto platform. Carto has several business oriented vertical products but does not have one oriented toward the humanities. While the platform is very good for building analytic applications, Carto does not provide some of the basic tools needed by the AMAP project such as data search, map composition, and data sharing [18].

- **MapStory** This open source map-based storytelling platform is oriented toward the display of temporal data and is capable of generating animations showing spatial change over time. MapStory is based on GeoNode, the same underlying system as WorldMap, and thus supports many of the same capabilities in terms of data curation and sharing. MapStory does not however support federated search or the discovery of data on other systems [19].

- **OmekaNeatline** Omeka is a humanities oriented content management system well suited to telling stories around web-based, humanities oriented content. It is being used to manage many kinds of collections including photos, documents, videos, and maps. Neatline is a plugin for Omeka's plugin architecture and allows users to tell complex spatial and temporal stories by overlaying various types of content on maps and associating them with dates and locations. OmekaNeatline does not include a data search capability [20].

- **Harvard WorldMap** WorldMap is a geospatial information sharing platform developed by the Center for Geographic Analysis (CGA) of Harvard University. It grew out of the CGA's experience building one-off web mapping tools for scholars, many of whom come from the social sciences and humanities. WorldMap allows scholars to create, curate, and share their materials online. WorldMap also supports data search and discovery by temporal extent, spatial extent, and keyword. Its database contains Chinese geographic information and maps on demographics, religion, transportation, urban studies, ethnicity and language, energy, environment, education, climate, public health, economics, and history, among many other subjects. To give an idea of the variety, there are maps of scholars in the Ming Dynasty, road networks of the Ming and Qing Dynasties, a geographical distribution of the Jinhua literati's social relations, distribution maps of Chinese temples in 1820, and distribution maps of defense posts in the Ming Dynasty [21].

WorldMap supported the core functions needed by the AMAP project, and being open source, can be adapted and installed inside the Zhejiang University firewall. For these reasons WorldMap was chosen as the best starting point for developing the AMAP system.

## 3. The Construction of the Chinese Academic Map Publishing Platform

The conception of this project and its unique approach started with the establishment of an international collaboration. Usually platform development starts with gathering user requirements followed by system design, proof of concept, prototyping, content creation, pre-release, and finally formal release. However, the construction of the AMAP system proceeded in a rather unconventional manner. Because the new system chose to adopt an existing platform (Harvard WorldMap, which is functional in the English environment) as the base system, and because user demand was urgent, content development started as soon as the base system was installed, while system customization and functional enhancements were ongoing. This process has proven to be efficient as the platform was made available for end users early on, and content accumulated quickly. However, this also created

challenges, as users often encountered unstable performance due to frequent code upgrades, and there was frequent debugging. See Section 4 below for more detail.

### 3.1. Establishing the Collaboration

Zhejiang University's Big Data and Humanities Academic Map Team was established in April 2017. The members are composed of faculty members from the School of Humanities, the Institute of Geographic Information Science, and the School of Computer Science and Technology. The team is affiliated with the Social Science Research Institute. It is focused on the construction of a geospatial database for Chinese cultural history data, and started conducting spatial analysis on the data. The team's goal is to build China's first culturally and historically oriented academic map publishing platform. The design of the platform aims to tightly integrate data and information on the backend with visualization and analysis capabilities on the front end.

Harvard University's digital humanities research is well recognized globally. Chinese cultural and historical databases built at Harvard include the Chinese Historical Geographic Information System (CHGIS), the Chinese Historical Biographical Database (CBDB), and the many China data and map collections stored in WorldMap. On 12–13 October 2017, President Wu Zhaohui of Zhejiang University led a delegation to visit Harvard. The team and the Center for Geographic Analysis signed a memorandum of understanding on the joint construction of the AMAP. According to the memorandum, the two sides would conduct cross-regional cooperation and jointly create an academic map publishing platform suitable for China's national conditions, presenting China's cultural history from a spatial dimension. Building on the foundation of Harvard's WorldMap, the team will reconfigure and improve it to form AMAP. To better meet the needs of Chinese users, the team will add Tianditu (天地图) as the Chinese base map, improve language translation for the user interface, add support for Chinese map projections, Chinese text encodings, and long text in spatial data attributes, among other enhancements.

### 3.2. Customizing the Current WorldMap for AMAP

In order for the WorldMap platform to meet the needs of the AMAP project, a number of enhancements were required. Because the source code for WorldMap is open, it was possible for the team to make the necessary changes. Below, we discuss each of the major enhancement areas and the way in which WorldMap was modified to support the new functions.

**Home page customization**. The original default WorldMap homepage was very simple but AMAP platform users needed to be able to easily determine the most popular and recent layers and maps, as well as have access to videos, including those describing how to use the system. The AMAP team therefore customized the default JavaScript homepage to include these new features.

**Multi-language user interface**. Because the main target audience for the AMAP system is scholars within China, it is critical that users have the option of a user interface written in Chinese. The WorldMap platform, by virtue of its underlying GeoNode content management system, can be configured to support a wide range of languages, including Chinese. The team examined all sections of the interface and created Chinese labeling wherever it was needed.

**Chinese language basemap**. There exist a number of remotely stored Chinese language basemaps, and it is important that the AMAP system is able to use them. WorldMap employs the OpenLayers JavaScript library for map display which provides a framework for bringing in new base layers, but some Chinese layers are not supported and require significant coding to enable them. The team found that some base layers are easier to add to the system than others. Depending on the technical underpinnings of a new base layer, it is not always possible to support all functions, such as automated map thumbnail generation and printing.

**Long text in layer attributes**. The AMAP project, being oriented towards the mapping of events in poetry and prose in Chinese, must be able to make extensive text content searchable by users. For this reason, the limitation in text length in WorldMap had to be eliminated. For layers that are created online in the system, there was no issue. However, for uploaded content, it is more complex given the

shapefile 256-character limitation. For this reason, it was necessary to add a new vector upload file type, GeoJSON, which has no such text length limitation.

**Feature level text search**. The AMAP project must support user search against spatial features which contain Chinese text, with the ability to highlight all matching features on a given layer or set of layers. This capability was developed for the AMAP system using the common query language (CQL) method of querying and highlighting a web map service (WMS) layer, which is supported by GeoServer, the underlying map renderer used by WorldMap.

**Bulk social media postings**. The AMAP project plans to use Chinese social media to promote content in the AMAP system, and to make the process of posting efficient. A capability was developed for generating a list of map or layer names with matching URLs ready for posting. This was developed as a JavaScript enhancement to the layer and map query pages in WorldMap.

**Mobile device search**. The AMAP platform must be accessible from mobile devices as well as from regular computers. To support this capability, a mobile map viewer was developed by the AMAP team in JavaScript that enables basic map navigation functions in the field.

### 3.3. Developing Content and Use Cases on the Platform

On 19 March 2018, AMAP was jointly established by the Big Data and Academic Map Innovation Team in the Social Science Research Institute of Zhejiang University and the Center for Geographic Analysis of Harvard University. The website (http://amap.zju.edu.cn) was officially launched, marking the birth of an academic map publishing platform designed to meet the needs of scholars in China.

In its first 18 months, more than 500 maps and 500,000 data sets were released on the platform with content, in addition to the humanities, covering the geosciences, agronomy, health, environment, transportation, climate, and meteorology. For example, the health category includes "The spatial and temporal distribution of infectious diseases in Hangzhou" while the food safety category includes "Food rumor distribution map", "Hangzhou Market Supervision Administration Food-related Rumor Video (with link)", "Fake Honey Video (with link)", "2011 Hangzhou Agricultural and Sideline Products Logistics Center Vegetable Pesticide Residue Survey", etc. The agronomy category includes "Zhejiang Silkworm Silk Weaving Proverbs Distribution Map" and "1993 Basic Situation of Zhejiang Agricultural Business Fields". The environmental category has "PM2.5 concentration detection". The climate and meteorological categories include "Zhu KeZhen track map" (竺可桢行迹图), "Zhejiang average temperature and annual differences". The economic and financial categories include "Zhejiang Population Growth Rate in the Late Qing Dynasty", "Zhejiang Banking (钱庄) Industry Status (Republic of China era)", "Zhejiang County Banks, Time of Establishment and Capital Composition", etc. The geosciences category includes "Land Use Performance Evaluation", "The First to the Sixth Releases of the Chinese Historical and Cultural Towns and Their Locational Index", "The First to the Sixth Releases of the Chinese Historical and Cultural Villages and Their Locational Index", "The First to the Fifth Releases of 5,825 Chinese Traditional Villages and Their Location Index", and many others.

At present, the platform has more maps in the humanities field than others. These include the distribution of poetry authors in "Complete Song Articles" (全宋文), "Complete Yuan Articles" (全宋文), "Complete Yuan Poetry" (全元诗) (Figure 1), "Complete Jin Yuan Ci" (全金元词) and "The Collection of Poems in the Dynasty" (列朝诗集小传), among other distribution data of character groups. The system also has data on opera authors of Ming and Qing Dynasties, female authors of Ming and Qing Dynasties, opera performers of the Qing Dynasty, writers of Jiangxi Province in the Song Dynasty, scholars from Jiangxi Province of the Song Dynasty, Model women mentioned in the history of the dynasties, writers from Yunnan Province of the Ming and Qing Dynasties, the ancient and modern figures of Zhejiang, females in the literary families around Taihu Lake in the Ming and Qing Dynasties, and distribution maps of ancient surnames (Wu, Zhou, Xu, Cha, Sheng, Jiang, Chen, Wang, Liu, Ye, Zhang). The geographical distribution data layers for non-human subjects include the "Siku Quanshu Catalog Summary" (四库全书总目提要) (Figure 2), "Zhejiang Literature Collection Catalog" (浙江集部著述总目), the geographical distribution of the literature collection catalogs in various provinces

(Jiangxi, Jiangsu, Hunan, Guangdong, Anhui, Sichuan, etc.) in the Qing Dynasty, and the subtitles of the "Continuation of Siku Quanshu" (续修四库全书). The maps of intangible cultural heritage include national and local intangible cultural heritage lists, shadow play distributions, and video data.

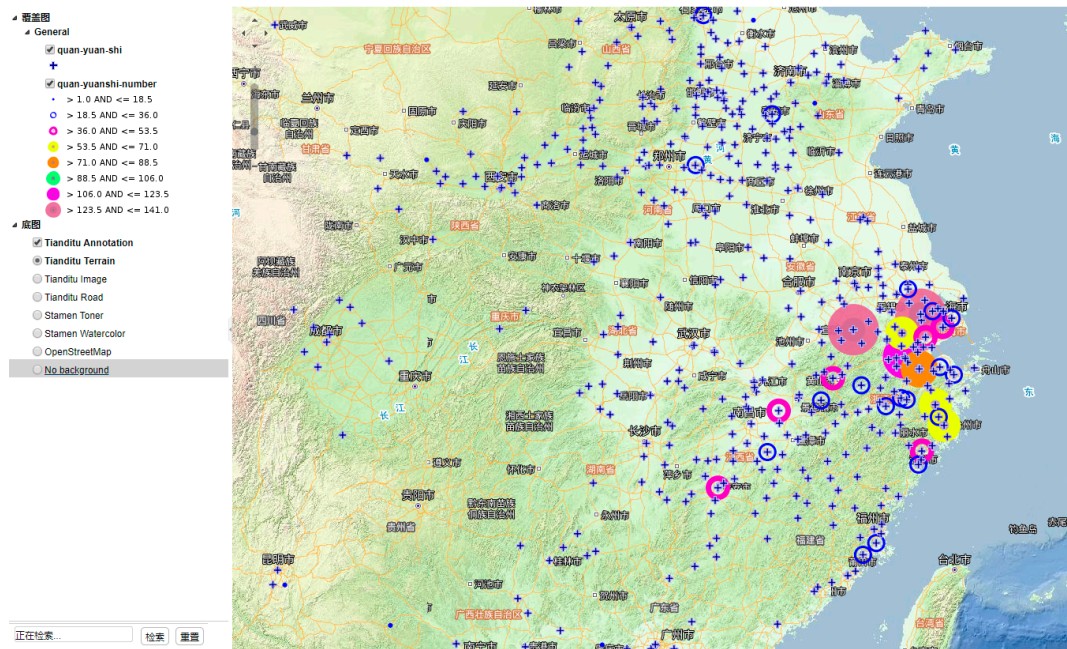

**Figure 1.** The distribution map of poetry authors in "All Yuan Poetry" (全元诗).

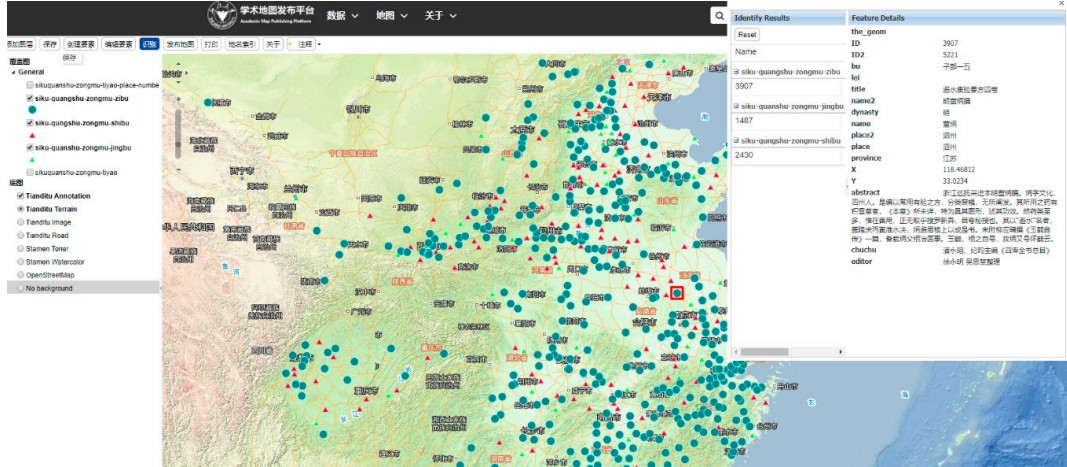

**Figure 2.** The distribution map of locations in "Siku Quanshu Catalog Summary" (四库全书总目提要).

Data about individual historical figures include Zhuge Liang 诸葛亮181–234), Lu Ji (陆机261–303), Wang Bo (王勃650–675) Chen Ziang (陈子昂659–700), Zhang Yue (张说667–731), Wang Wei (王维701–761), Bai Xingjian (白行简776–828), Du Mu (杜枚803–852), Feng Yansi (冯延巳903–852), Yan Shu (晏殊991–1055), Shi Chengxun (释成寻1011–1081), Su Shi (苏轼1037–1101, Figure 3), Hong Mai (洪迈1123–1202), Lu You (陆游1125–1209), Zhu Xi (朱熹1130–1200), Xin Qiji (辛弃疾1140–1207), Ouyang Xuan (欧阳玄1283–1357), Wu Shidao (吴师道1283–1357), Shi Naian (施耐庵Approximately 1296–1370), Song Lian (宋濂1310–1381), Liu Ji (刘基1311–1375), Gao Qi (高启1336–1373), Xie Jin (解缙1396–1415), Shen Zhou (沈周1427–1509), Chen Xianzhang (陈献章1428–1500), Li Dongyang (李东阳1447–1516), Wang Ao (王鏊1450–1524), Yang Yiqing (杨一清1454–1530), Zhu Yunming (祝允明146–1526), Wang JiuSi (王九思1468–1551), Tang Yin (唐寅1470–1523), Wen Zhengming (文征明1470–1559), Wang Yangming (王阳明1472–1528, Figure 4), Xia Yan (夏言1482–1548), Gui Youguang (归有光1506–1571), Wang

Shenzhong (王慎中1509–1559), Li Panlong (李攀龙1514–1570), Shen Mingchen (沈明臣1518–1596), Xu Wei (徐渭1521–1593), Wang Daokun (汪道昆15251593), Wang Shizhen (王世贞1526–1590), Li Zhi (李贽1527–1602), Tang Xianzu (汤显祖1550–1616), Feng Qi (冯琦1558–1603), Ye Xianzu (叶宪祖1566–1641), Yuan Hongdao (袁宏道1568–1610), Feng Menglong (冯梦龙1574–1646), Qian Qianyi (钱谦益1582–1664), Zhang Dai (张岱1597–1685), Chen Zilong (陈子龙1608–1647), Huang Zongxi (黄宗羲1610–1695), Fang Yizhi (方以智1611–1680), Li Yu (李渔1611–1680), Gu Yanwu (顾炎武1613–1682), You Dong (尤侗1618–1704), Hou Fangyu (侯方域1618–1704), Zhu Yizun (朱彝尊1629–1709), Qu Dajun (屈大均1630–1696), Wang Shizhen (王士1634–1711), Kong Shangren (孔尚任1648–1718), Quan Zuwang (全祖望1705–1755), Zhang Tingyu (张廷玉1672–1755), Yao Nai (姚鼐1731–1815), Nalan Rongruo (纳兰容若1655–1685), Zheng Banqiao (郑板桥1693–1765), Yuan Mei (袁枚1716–1797), Dai Zhen (戴震1724–1777), Gong Zizhen (龚自珍1792–1841), Zeng Guofan (曾国藩1811–1872), Zhang Yuanji (张元济1867–1959), Lu Xun (鲁迅1881–1936), Xia mianzun (夏尊1886–1946), Mao Dun (茅盾1896–1981), Xu Zhimo (徐志摩1896–1931), Yu Dafu (郁达夫1896–1945), Sun Kaidi (孙楷第1898–1986), Yu Pingbo (俞平伯1900–1990), Feng Xuefeng (冯雪峰1903–1976), Lin Huiyin (林徽因1904–1955), Li Jianwu (李健吾1906–1982), and Zhang Ailing (张爱玲1920–1995), et al.

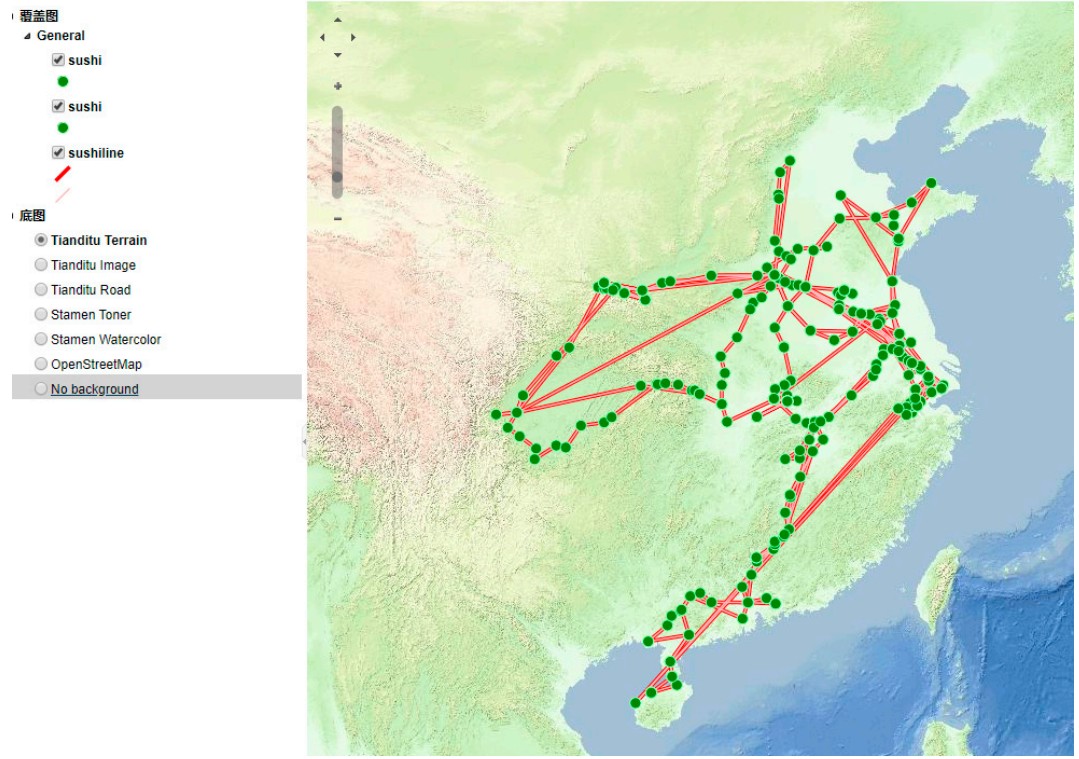

**Figure 3.** Movement tracks of Su Shi (苏轼 1037–1101).

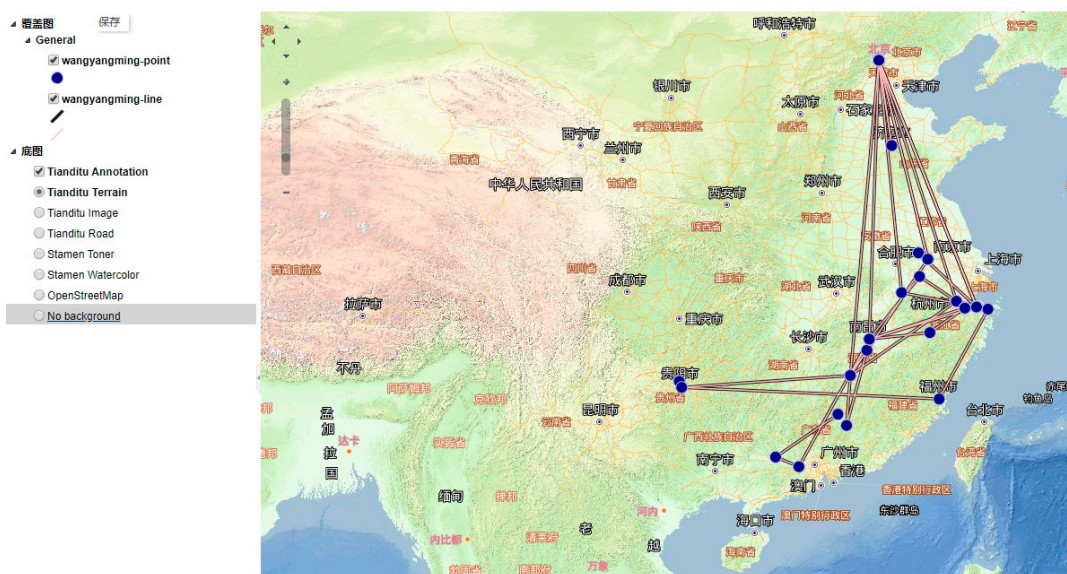

**Figure 4.** Movement tracks of Wang Yangming (王阳明 1472–1528).

*3.4. Functions of the Platform*

The AMAP platform provides users with functions such as publishing, editing, searching, viewing, query by location, and sharing. Each area is briefly described below.

- **Publish a map**: If the user has built an Excel data table with latitude and longitude, she may wish to publish it on AMAP. A basic procedure for doing this is to save the Excel file in CSV format, load it to QGIS, export it to the shapefile format from QGIS (Shapefiles contain multiple files with the suffixes dbf, prj, shp, and shx), and then upload the shapefile (multiple files) to the AMAP platform to form a layer. The user may click on the layer to enter the layer page, click the "Create a map" button, select Tiantidu as the base map, and give the created map a name and save it. In this way, a map may be created and published.

- **Edit a map**: The AMAP platform provides editing capabilities for uploaded layers and for created maps. Editing functions include setting thumbnails, defining symbology for geographic features, and controlling access permission. For example, when a user wishes to set a layer's access permission, the platform provides several settings: "Who can view it?", "Who can download it?", "Who can change the metadata?", "Who can edit the data in this layer?", "Who can edit the symbology for the layer?", and "Who can manage it?".

- **Search for maps**: For published maps, the platform provides options to search by map name and layer name. For example, if a researcher would like to search for keywords with "Hangzhou" in the abstract of the map, she may find "Distribution of Zhejiang Weisuo in the Ming Dynasty", "Index of Authors of Ming Dynasty Female Authors", "Zhejiang Ming Dynasty Characters", and "Map of Ming Dynasty Stations and Roads", among others. If she entered "Zhejiang", all maps with "Zhejiang" in the map name or abstract will be displayed.

- **View a map**: After finding a map, the researcher can preview the map on the "View Map" page. The specific data hidden in the map is not visible in preview. Once the user enables the "Identify" button, clicking on geographic features on the map will bring up attribute information.

- **Find a location**: The platform provides the ability to search for a location. After opening a map, on the menu bar there is an icon for "Place Name Index" or "Gazetteer search". In this tool one can enter a place name, such as "Ningbo", and all place names containing "Ningbo" will appear. If one clicks on the layer to go to the map associated with the layer, the user may get the information needed. If the user wanted to locate the place, the user may double-click the place name and a popup window containing the candidate locations for this place name will appear on the current

map. The user may then pick one to determine the location of the place name. This "Gazetteer search" can be used both to find places and to locate other key fields such as "person name", "book name", and "era" if they are selected as gazetteer fields in the layer's metadata page.

- **Share a map**: Users may share URL links to their own maps or maps posted by others. The platform provides more than 10 kinds of social media sharing methods, including: Email, Facebook, Twitter, Google+, WeChat, QQ, QQ Space, Tencent Weibo, Sina Weibo, Renren.com, Tieba.Baidu.com, and Douban.com.

## 4. Challenges and Opportunities

The development of AMAP is an ongoing endeavor by the teams at Zhejiang University and Harvard University. In less than two years, the two groups have overcome a range of challenges which we describe below.

### 4.1. Internet Environment

The Internet has been expanding rapidly in China since the 1990s. Early on, there was a brief period of free communication with the world outside China, which was soon followed by tightening government controls. This domestication of the Chinese internet has been a multi-directional process involving multiple social actors, complex flows and interactions, and polyvalent and ambivalent outcomes. In just a little over two decades, the Chinese government has perfected a system of Internet control, including monitoring, blocking or filtering of information from outside China, and the censoring of information inside. A complex system of IP banning, DNS spoofing and redirection, URL filtering, and packet forging, popularly dubbed the "Great Firewall" was erected as a virtual boundary, selectively separating Chinese cyberspace from the outside [22].

The Chinese national internet firewall blocks a growing inventory of international web services including Google base maps. There is also a severe time delay when accessing those international services which are allowed from inside China. Individual organizations in China, such as universities often implement an even stricter network security control, forbidding the consumption of any international web services, and blocking all access from non-Chinese IP addresses. China's internet is now like a huge intranet, containing many small intranets inside it [23]. This design has obvious advantages in terms of network security, but it also causes headaches for international collaborations which rely on shared web platforms and collaborative code development to be successful.

The development of the AMAP system was hindered by this restrictive internet environment in a number of ways. One direct impact was the inability of AMAP to access remote mapping layers. For example, an AMAP instance running on the Zhejiang University server can only accept data layers stored in its local database and cannot take advantage of the global web map service registry in WorldMap [24]. This limitation forced the development team to focus on data layers contributed by users in China exclusively.

Because international developers cannot have direct access to the AMAP server on the Zhejiang University intranet, work on system configuration, performance diagnosis, and debugging had to be performed by the Zhejiang University team without direct outside support. Issues, testing, as well as remediation suggestions and their outcomes were communicated between the two teams asynchronously. The difference in time zones made the process much slower than onsite operations would have been, with a typical dialog cycle taking two days. Often, a test which could have been done in minutes onsite would take days, sometimes even weeks, to manage between the two teams. To overcome the ambiguity of verbal descriptions, the teams used screenshots and videos to document issues and solutions, which though taking more time to create, results in more accurate documentation.

The blockade of popular international base maps required AMAP to replace WorldMap base maps such as Google Maps and Open Street Maps with Chinese base maps such as Tiantidu [25]. Because Tianditu is not used in international applications outside of China (see the discussion in the "Map projection and datum" section below), the Harvard team and the international open source

developer community for Geonode (the core component of WorldMap) has little experience with Tianditu. Once again, the Zhejiang University team was on its own handling issues related to Tianditu base map integration.

### 4.2. Character Encoding

Spatial data in vector format usually contains attributes, or tables of values which are associated with geographic features (points, lines or polygons). Each cell in such a table contains a sequence of characters representing a number, a text string, a date, a URL, etc. In different languages, these characters may be composed of different letters of the alphabet, different punctuation, etc. When stored in a computer, these values are converted to a sequence of bytes, or numeric values. Sometimes more than one byte is used to represent a single character. The binary key used to translate between a sequence of bytes and their corresponding character is known as character encoding.

There are many different encoding systems, some for a specific language, such as the Chinese standard Guojia Biaozhun Kuozhan (GBK) for simplified Chinese characters, and others which support multiple languages, such as Unicode Transformation Format with 8 bytes (UTF8). Most software systems use a default encoding that works for the most common languages used by the intended user community. AMAP allows the user to inform the system at upload time of the correct encoding, enabling the data to be displayed correctly.

WorldMap, as a system intended for the international user community, uses UTF8 as the default encoding. However, most Chinese users are more familiar with operating systems that use language settings which default to the GBK encoding. Even though WorldMap can also handle GBK encoding, it is not the default, and thus requires users to declare it specifically when they upload their data. Alternatively, users may convert their data to UTF8 so that when they upload it to WorldMap, the system will handle it correctly by default. This seems to be an easy to understand process, however, even with user training and help documents, encoding mismatch is one of the most common user problems. To better address this challenge, the team will revise training materials to give more emphasis on character encoding.

### 4.3. Map Projection and Datum

A map projection is a systematic transformation of geographic coordinates that converts the coordinates of locations recorded as latitudes and longitudes on the three-dimensional earth surface, into coordinates on the Cartesian plane of a map. The shape of the earth is modeled in many ways, including as a sphere, an ellipsoid, or more accurately, as an irregular 3-D object, using a geodetic datum. A datum is a reference system which defines the shape of an ellipsoid which approximates the earth's shape, and combined with a projection system, allows people to reliably share information about locations on the surface of the earth.

Because of the irregular shape of the earth and the transformation from a 3-D surface to a 2-D plane, all map projections, regardless of the datum referenced, introduce locational distortion. Typically, when creating a map, a map maker selects the datum and projection that best preserves locational accuracy and attempts minimize distortion for a given region. However, for security concerns, a country may develop a special datum that purposely distorts geographic coordinates relative to common datums such as WGS-84. Maps created in these countries may be legally required to use a map projection that references such a datum. China's GCJ-02 datum is such a geodetic datum. It was formulated by the Chinese State Bureau of Surveying and Mapping, which developed a topographic map non-linear confidentiality algorithm based on WGS-84, the internationally recognized reference coordinate system used by the global positioning system.

A marker with GCJ-02 coordinates will be displayed at the corresponding location on a GCJ-02 map. However, if placed on a WGS-84 map, the coordinate offsets can result in up to a few hundred meters error. Similarly, a WGS-84 marker (such as an untampered GPS location) when placed on a GCJ-02 map, will also show up at a seemingly random location somewhere within a radius of a

few dozen to a few hundred meters. There are official transformation tools for converting WGS-84 coordinates into GCJ-02 coordinates, but there is no official tool for the reverse transformation [26].

Web map services meeting this Chinese government mandate must reference the GCJ-02 datum, or use another government approved datum which provides additional distortion, such as those used in commercial map services provided by the Chinese companies Baidu and Amap. Some international map service providers complied with China's requirements, such as Google; others did not, such as OpenStreetMap. The discrepancy between GCJ-02 and WGS-84 is clearly visible on the BBBike web map comparison application [27].

AMAP's primary user community is in China, therefore it is required to use Chinese base maps. The default base map in AMAP is Tianditu. When users upload spatial data layers created in China referenced to GCJ-02 base maps or using Chinese government approved GPS devices, the features will line up with the Tianditu base map, preserving the relative locational accuracy. AMAP also provides an OpenStreetMap base map as an alternative, to support data referenced to the WGS-84 datum, in which case the user may manually switch the base map to OpenStreetMap. For most AMAP users however, the mapping is at a small enough scale that a shift of a few hundred meters does not matter, thus this problem will not be of great concern for them. Many spatial datasets created in earlier years in China are referenced to the Xian 1980 datum, which can be transformed to GCJ-02 or WGS-84 using popular GIS tools such as ArcMap or QGIS.

### 4.4. Software Development Culture

Software engineers from China have played an increasingly important role in the open source sector [28]. The total number of GitHub users and contributors from China has been growing rapidly, promoting China's ranking among world countries [29]. However, open source is a relatively new practice in China with many challenges [30]. Despite being ranked among the top three countries by total number of open source contributors, China has not made it to the top 25 countries by number of contributors per capita [31].

Even though software code has become a virtually universal language, human language still plays an important role in software collaborations. Living in a non-English-speaking society, Chinese software engineers must first overcome the language barrier when collaborating with the global community [32]. Perhaps more important than the human language difference, however, is the difference in the culture of communication. In many Chinese organizations, employees are used to hierarchical communication chains within the organization and one-to-one contacts across organizations. Compared with instant team-wide messaging on Slack and other channels, the hierarchical style is more effective in carrying out a well-defined and tightly managed project, but it is less effective in harnessing creative ideas in a loosely coordinated group, which most of the open source communities are. The AMAP project experienced this cultural difference in communication in the earlier stages of the project and it took some time for both teams to adjust their communication styles to accommodate one another.

Another major challenge for the AMAP project is the small number of Chinese language users in the international GeoNode open source community. Other major efforts around the world that contribute to GeoNode are English-language based. This holds true even for those projects that are based in Europe. Therefore, when new code is introduced to GeoNode, there are few Chinese pilot users to test the Chinese-related functions. Bugs and regressions that are specific to Chinese language encoding, map projection, or data structure remain undiscovered in new releases of GeoNode due to lack of testing by Chinese users. This situation led to a number of setbacks to the AMAP project which occurred after version upgrades.

One of the initial objectives of the AMAP project was to collaboratively develop a software platform, knowing the teams are from different cultures, with the hope that in the process, each team would learn technical skills from the other, while improving their ability to solve technical problems in cross-cultural situations. The two teams have now set off to collaboratively build AMAP, a Chinese-centric platform for humanities research. The preliminary results are encouraging. The

platform is being widely used even while under construction, despite the glitches pre-release platforms commonly have. As the Zhejiang University team becomes more familiar with the GeoNode-based AMAP system, and the open source community it is part of, there will be a transition of responsibility from the Harvard team to the Zhejiang team for future enhancements and design changes.

## 5. Future Perspectives

We believe that the establishment of the AMAP platform is important for the following reasons:

- The platform is China's first multi-purpose geographic information platform that supports the publishing of geographic information data on any historical subject matter.
- The platform supports the creation of data on a system run within China rather than on a server run within another country as has often been the case in the past.
- Contents on the platform may serve as reference for government decision-making and social services, in addition to scientific research.
- The platform promotes the use and sharing of digital spatial data from heterogeneous sources within an academic context.
- The platform is useful for primary and secondary school students who wish to learn about geographic conditions in China at national, provincial, and local scales.
- There is potential for the platform to support the development of smart tourism services.

The overarching objective of AMAP is to build a platform for creating, organizing, and sharing historical datasets to support qualitative and quantitative work in the humanities in China. In addition, we believe there is a potential for this system to have an impact outside the academy in support for government decision-making, especially with regard to planning for social services and tourism. To achieve these goals, we envision building data holdings in the following key areas:

- Map the movements of all documented historical figures. There are tens of thousands of such historical figures from before the Qin Dynasty up to the present.
- Map the distribution of Chinese writers throughout history, including large-scale collections, female writers of all dynasties, and writers of various styles.
- Map the locations mentioned in historical literatures, such as "CompleteTang Dynasty Poems" (全唐诗), "Complete Song Dynasty Poems" (全宋诗), "Complete Yuan Dynasty Poems" (全元诗), "CompleteSong Dynasty Lyrics" (全宋词), "Complete Jin and Yuan Dynasty Lyrics" (全金元词), "CompleteTang Dynasty Essays" (全唐文), "Complete Song Dynasty Essays" (全宋文), "CompleteYuan Dynasty Essays" (全元文), and poems and essays of Ming and Qing Dynasties.
- Develop data as well as video and audio files for cultural heritage sites, scenic parks, former residence of celebrities, temples and cultural gathering places.
- Map the catalog of historical literature from Zhejiang Province.
- Develop a database for Chinese local historic gazetteers, including data about officials, imperial examinations, famous characters, arts and literature (writings).
- Develop data for key villages and towns.
- Develop data for artists.
- Develop data for intangible cultural heritage.

## 6. Conclusions

The goal of the AMAP system is to provide for the first time, a system that humanities scholars in China can use to create, organize, discover, analyze, visualize, and share research content, and benefit from the ability to organize and visualize that content by geographic space as well as by time. The platform initially provided most of its content from the humanities, but has since expanded to other subject areas, including social sciences, environmental science, public health, business, tourism, public service, government administration, and others. The system has been used to support teaching at the

college level, and has the potential to be adopted for K-12 classrooms. The system also holds promise for addressing another important area: improving the availability of scholarly data within China and facilitating the sharing of that data within China and globally.

As the initial development phase of the system nears completion, and the project enters a new phase of functional enhancement and content expansion, AMAP will face new challenges. There will remain the need for a strong software engineering team, knowledgeable about the unique technical environment in China and the specific needs of Chinese scholars, but this same team must also remain well connected to the global open source GeoNode developer community. In this regard, it will be critical to find a way to facilitate effective knowledge transfer between the current collaborating team members and participating graduate students as they progress in their graduate program. In the longer term, there may also be a role for a private companies based in China to help maintain and grow the system.

As an open source system like AMAP evolves, it is especially important to find a way to balance the short term need of adding new functions quickly, (perhaps forking the code to bypass the lengthy process of getting buy-in from the community and checking in changes to the core code base), with the long term need to remain synchronized with the latest version of the code base. Once a system is forked, it can become costly to migrate back to the main code base again, especially if the fork has been allowed to persist for months or years. If the system remains unsynchronized with the main code base, the project can lose the benefit of contributions from the global open source community [33].

As content accumulates and the number of users increase, maintaining system stability and performance will become more challenging. Security, backup and recovery, archiving, and preservation, for the system code, its digital contents, as well as the published maps and data as permanent links, will demand significant attention and resources. The future of AMAP and its sustained growth will therefore depend largely on the availability of resources, both in terms of hardware and people.

As an academic research project, the development of AMAP has achieved its initial goals. In addition to the platform as a product and the assembled content, the knowledge gained by the team and the lessons learned during the system's development, especially as it relates to the international nature of the collaboration around an open source project, has been valuable. Our hope is that this knowledge, being well documented, will benefit other groups that engage in similar collaborations.

**Author Contributions:** Conceptualization, Yongming Xu, Benjamin Lewis and Weihe Wendy Guan; Data curation, Yongming Xu; Formal analysis, Benjamin Lewis; Funding acquisition, Yongming Xu and Weihe Wendy Guan; Investigation, Benjamin Lewis and Weihe Wendy Guan; Methodology, Yongming Xu and Benjamin Lewis; Project administration, Yongming Xu and Weihe Wendy Guan; Resources, Yongming Xu; Software, Benjamin Lewis; Supervision, Weihe Wendy Guan; Validation, Yongming Xu and Benjamin Lewis; Visualization, Yongming Xu and Benjamin Lewis; Writing—original draft, Yongming Xu, Benjamin Lewis and Weihe Wendy Guan; Writing—review & editing, Yongming Xu, Benjamin Lewis and Weihe Wendy Guan.

**Funding:** This project is funded by Zhejiang University and Harvard University. The platform benefited from other projects contributing to the GeoNode code base, especially the Secondary Cities project funded through NSF #1841403.

**Acknowledgments:** The authors wish to thank all team members who contributed to this project both in terms of system code and in platform content development, especially Paolo Corti, Lijun Wang, Yao Liu and Feng Zhang.

**Conflicts of Interest:** The funders had no role in the design of the study; in the collection, analyses, or interpretation of data; in the writing of the manuscript, or in the decision to publish the results.

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
