# Peer review of "Developing the Chinese Academic Map Publishing Platform"

_ijgi, doi:10.3390/ijgi8120567_

Round 1
Reviewer 1 Report
This is a very important paper describing the first open access geospatial platform in China. It explains in clear language the need for the platform in the context of Chinese history and literature as well as its value for spatial datasets on contemporary China. The platform is based on WorldMap.harvard.edu. The paper explains what needed to be done to make this system work for Chinese users. It frankly discusses the challenges of international collaboration across the Chinese firewall. This article will be an important historical record of a platform that may become, given the Chinese population and its level of education, the most used mapping platform in the world.
A final review of the text can remove the very few minor problems I encountered.
Because reference to most texts and persons include Chinese characters, can this possibly be done for the datasets mentioned on lines 328-345?
Author Response
Thanks for the constructive review comments. We have revised the manuscript thoroughly to address this and all other review comments, as well as polished the language and corrected formatting errors throughout the paper.
Most of the geographic data layers in the AMAP database contain Chinese characters in theirs attribute tables. The system can accept Chinese text in multiple encoding formats, such as UTF8 or GBK, provided that the metadata record correctly identifies the encoding used. Chinese text may also be used in layer names and map names. One remaining limitation is that field names in the attribute tables must be ASCII text, which is an area for improvement in the future.
Reviewer 2 Report
The article raises the development of the Chinese Academic Map Publishing platform (AMAP).
Abstract: The abstract should be written as a 250-word mini-article that contains the research approach, the methodology, the results and the main conclusions. In a JCR publication, phrases such as: "We gained much experience in this multilevel cooperation between cultures, disciplines, universities, and technology implementations, and learned difficult but important lessons" are far from a scientific article.
Introduction: The introduction should, first of all, describe the purpose and why of the research carried out, and not begin by listing the parts of it.
Point 2. An Overview of Existing Platforms for Historical GIS and Digital Humanities: The authors provide much information but little relevant to the purpose of the article.
Point 3: The Construction of the Chinese Academic Map Publishing Platform: the authors describe, without going into much detail, how more than 500 maps and more than 500,000 data sets have been released on the platform, focusing their theme on humanities fields, as the authors claim. The functions described are those known in other types of geographic information servers such as GIS or spatial data infrastructures: publish a map, edition of maps, viewing maps ...
Conclusions: In the conclusions there are no results of the implementation of the AMAP system. It does not provide data on its acceptance among users, nor on its usability, for example. There are no results in themselves on which to conclude. The conclusions are a succession of the authors' statements, without supporting data.
References: It is striking that, in a scientific article, we find hardly any reference of other authors in JCR publications. References are a succession of websites.
Since the abstract I have missed the typical structure of a work of a scientific journal: Introduction, methodology, data analysis, results and conclusions. Then, in reading the article, I have verified that the authors report the construction of the Chinese Academic Map Publishing platform (AMAP) but that they do not provide a specific methodology on the tool developed, in which they provide data on users, developers, teachers, implementation, operation ... It is a description of the steps taken to implement the Chinese Academic Map Publishing platform (AMAP).
From the point of view of originality, the construction of this platform is not too original, since the authors detail things like home page customization, multi-language user interface ... The platform, as the authors claim, has more maps in the humanities field than others, which brings it closer to journals of humanities and / or education. The article does not provide results that provide an advance in current knowledge.
From the point of view of significance: There is no interpretation of results, since there are none. The conclusions, therefore, are not justified and supported by the results. Likewise, although the introduction refers to the construction of the platform, the hypotheses are not clearly identified.
For this reason, beyond detailing the strengths and weaknesses of the article, I consider it closer to more generalist humanities journals, and not for the International Journal of Geo-Information.
Author Response
Thank you very much for the detailed and thoughtful review comments. We have revised the manuscript thoroughly to address each of your review comments.
We rewrote the Abstract to mention specific lessons learned.
We rewrote the Introduction to first describe the purpose of the research.
Point 2. In An Overview of Existing Platforms for Historical GIS and Digital Humanities, we consider the information relevant to the purpose of the article, because these systems are online mapping systems useful to humanities research. We needed to review their capabilities and limitations to determine if any of them could meet the needs for our intended use. Only when knowing what is available and what is missing, we can conclude that a new system is indeed needed, and it can be built based on one of the existing platforms. This is a fundamental step in this research project.
Point 3. The purpose of this research and this paper is not about how to create the content in these maps and datasets. It is about how to make them available to collaborators and the public. It is true that the mapping functions of the AMAP platform is no more than common GIS software packages. The unique value of AMAP, and this research project, is to make such GIS and mapping tools available to the Chinese research community via a browser, allowing them to upload their own datasets, compose maps, and share them with whomever they want, without having to purchase, install and maintain any GIS software.
Conclusion: At this time, the result of the implementation of the AMAP system is the system itself and its contents (maps and datasets), as well as the lessons learned in building the system which this paper shared. Even though the system has been in active use since early 2018, it is still under development, thus not mature for usability testing at this time.
References: We have enhanced citation with several additional research articles while kept the websites in the references. We understand that it may be unconventional for humanity research to cite websites as references, however, due to the nature of this research, websites are the best sources for the systems we evaluated. Open source mapping platform development is a research area that evolves quickly. Journal articles or books may become outdated soon after (or even before) they come through the traditional publication process.
This paper indeed is a description of the steps taken to implement AMAP. We consider these steps the methodology necessary for anyone who may attempt to build a similar system. There is original code written for AMAP, but that is not what this paper is intended to report. The code base for the system is available on GitHub, thus there is no need to describe it by words here. This research is not a study of the humanities, it is a study of how to enable Chinese study of humanities with a collaborative mapping platform. The result is the platform itself and the lessons learned in building it. There is no hypothesis involved in this study. We knew we could build a system, it is just a matter of how, which this paper reported.
Reviewer 3 Report
Dear authors
I reviewed the paper Developing the Chinese Academic Map Publishing Platform.
The paper presents the Chinese Academic Map Publishing Platform (AMAP) project. AMAP aims to facilitate the accumulation of spatiotemporal datasets, support multi-faceted queries, and provide integrated visualization. The software is built on Harvard's WorldMap.
The paper is specific and the research design is not characteristic of publication in this journal.
Given the importance of the AMAP project, I recommend it for publication in its current form as a review paper.
Best regards
Author Response
Thank you for the thoughtful comment. We fully respect the editors’ decision on which category this paper may be most suitable to publish in.
Reviewer 4 Report
The paper documents an academic research project and it demonstrates the use of open source software to develop an academic map publishing platform for historic and cultural heritage data of China. It also demonstrates the challenges met by the team during the development phase in an international collaboration. The paper is well written, illustrates the practical work done, promotes the use of open source software in academic environment and may be used as a reference for future similar developments.
Introduction and references should be improved to include more research articles, rather than mostly online resources.
Considering text formatting there are some paragraphs that are not aligned. Also, there are extra white spaces between words throughout the text.
Author Response
Thank you for the constructive review comments. We have revised the manuscript thoroughly, especially the Introduction section, to address this and all other review comments, as well as polished the language and corrected formatting errors throughout the paper. Citation is enhanced with several additional research articles.
Round 2
Reviewer 2 Report
The changes that the authors have made have not solved the deficiencies that, from my point of view, I had detected. I believe that the article still needs additional experiments that provide results. I am not sure that the paper's approach is very original for an IJGI reader. The GIS are already very implanted already and the creation of one more does not believe that it contributes much to the scientific knowledge, although the work that the authors have done is remarkable.
I believe, therefore, that the article could succeed in another journal.
Author Response
Thank you for your time, and for sharing your comments and suggestions. We fully respect your judgment.